# Successor Representations Enable Emergent Compositional Instruction Following

## Abstract

Effective task representations should facilitate compositionality, such that after learning a variety of basic tasks, an agent can perform compound tasks consisting of multiple steps simply by composing the representations of the constituent steps together. While this is conceptually simple and appealing, it is not clear how to automatically learn representations that enable this sort of compositionality. We show that learning to associate the representations of current and future states with a temporal alignment loss can improve compositional generalization, even in the absence of any explicit subtask planning or reinforcement learning. This approach is able to generalize to novel composite tasks specified as goal images or language instructions, without assuming any additional reward supervision or explicit subtask planning. We evaluate our approach across diverse tabletop robotic manipulation tasks, showing substantial improvements for tasks specified with either language or goal images.

## 1 Introduction

Compositionality is a core aspect of intelligent behavior, describing the ability to sequence previously learned capabilities and solve new tasks (Lashley, 1951). In domains involving long-horizon decision-making like robotics, various learning approaches have been proposed to enable this property, including hierarchical learning (Kulkarni et al., 2016), explicit subtask planning (Schrittwieser et al., 2021; Fang et al., 2022b; Ahn et al., 2022), and dynamic-programming-based "stitching" (Ghugare et al., 2023; Kostrikov et al., 2022). In practice, these techniques are often unstable and/or data-inefficient in real-world robotics settings, making them difficult to scale (Laidlaw et al., 2024).

By contrast, biological learners are adept at quickly composing behaviors to reach new goals (Lashley, 1951). Possible explanations for these capabilities have been proposed, including the ability to perform transitive inference (Ciranka et al., 2022), learn successor representations and causal models (Dayan, 1993b; Gopnik et al., 2017), and plan with world models (Vikbladh et al., 2019). In common among these theories is the idea of learning structured representations of the world, which inference about which actions will lead to certain goals.

How might these concepts translate to algorithms for robot learning? In this work, we study how adding an auxiliary successor representation learning objective affects compositional behavior in a real-world tabletop manipulation setting. We show that learning this representation structure improves the ability of the robot to perform long-horizon, compositionally-new tasks, specified either through goal images or natural language instructions. Perhaps surprisingly, we found that this temporal alignment does not need to be used for training the policy or test-time inference, as long as it is used as an auxiliary loss over the same representations used for the tasks. An example of this can be seen in Fig. 1.

We evaluate our method, **T**emporal **R**epresentation **A**lignment (TRA), on a set of challenging multi-step manipulation tasks in the BridgeData setup (Walke et al., 2023). These tasks specifically test the compositional capabilities of the robot policies: as a whole, the tasks are out-of-distribution, but each distinct subtask can be described through a goal image that lies in the training distribution. Adding a simple time-contrastive alignment loss improves compositional performance on these tasks by >40% across 13 tasks in 4 scenes.

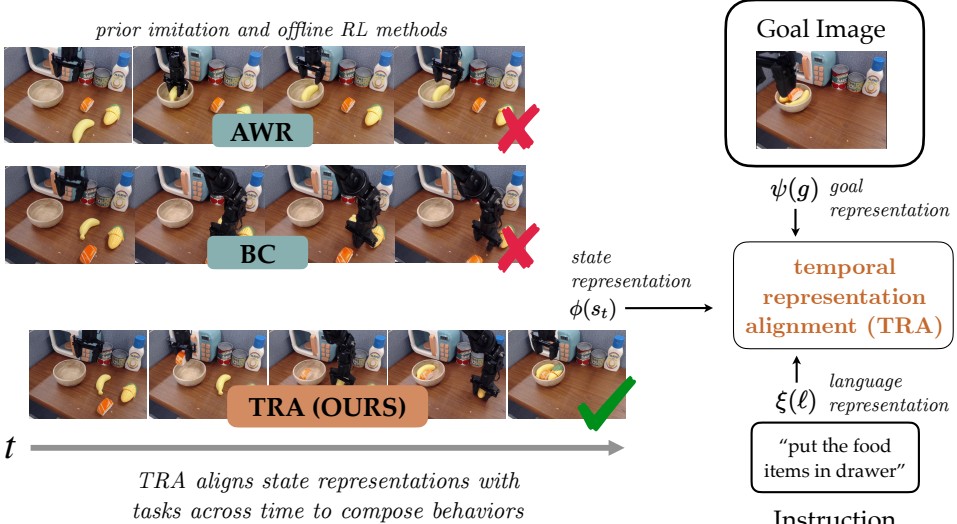

Figure 1: Example rollouts of a task with TRA and GCBC to put all food items in the bowl. While TRA can implicitly decompose the task into steps and execute them one by one, GCBC is unable to do that and fails to ground to any relevant objects. GCBC+AWR on the other hand only grounds one object, failing to display any compositionality.

## 2 RELATED WORK

Our approach builds upon prior work on goal- and language-conditioned control, focusing particularly on the problem of compositional generalization.

**Robot manipulation with language and goals.** Recent improvements in robot learning datasets have enabled the development of robot policies that can be commanded with image goals and language instructions (Ahn et al., 2022; Walke et al., 2023; Shridhar et al., 2021). These policies can be trained with goal- and language-conditioned imitation learning from human demonstrations (Chowdhery et al., 2023; Jiang et al., 2023; Lynch and Sermanet, 2021; Lynch et al., 2023; Brohan et al., 2023), reinforcement learning (Chebotar et al., 2023; Chen et al., 2021), or other forms of supervision (Bobu et al., 2023; Cui et al., 2023). When being trained to reach goals, methods can additionally use hindsight relabeling (Andrychowicz et al., 2017; Kaelbling, 1993) to improve performance (Walke et al., 2023; Myers et al., 2023; Dehaene et al., 2022; Ding et al., 2019). Our work shows how the benefits of goal-conditioned and language-conditioned supervised learning can be combined with temporal representation alignment to enable compositionality that would otherwise require planning or reinforcement learning.

**Compositional generalization in sequential decision making.** In the context of decision making, compositional generalization refers to the ability to generalize to new behaviors that are composed of known sub-behaviors (Rubino et al., 2023; Steedman, 2004). Biological learning systems show strong compositional generalization abilities (Ciranka et al., 2022; Dehaene et al., 2022; Dickins, 2011; Lake et al., 2019), and recent work has explored how similar capabilities can be achieved in artificial systems (Akyürek et al., 2021; Ito et al., 2022; Lewis et al., 2024). In the context of policy learning, exploiting the compositionality of the behaviors can lead to generalization to unseen and temporarily extended tasks (Ghugare et al., 2023; Kumar et al., 2022; Fang et al., 2019; 2022b; Mandlekar et al., 2021; Nasiriany et al., 2019). Hierarchical and planning-based approaches also aim to enable compositional behavior by explicitly partitioning a task into its components (Fang et al., 2022a; Myers et al., 2024a; Zhang et al., 2022; Park et al., 2023). With improvements in vision-language models (VLMs), many recent works have explored using a pre-trained VLM to decompose a task into subtasks that are more attainable for the low-level manipulation policy (Ahn et al., 2022; Attarian et al., 2022; Belkhale et al., 2024; Kwon et al., 2023; Myers et al., 2024a; Singh et al., 2023; Zhang et al., 2023). These approaches are limited by the need for robust pre-trained models that can be fine-tuned and prompted for embodied tasks. Our contribution is to show compositional properties can be achieved *without* any explicit hierarchical structure or planning, by learning a structured representation through time-contrastive representation alignment.

**Representation learning for states and tasks.** State and task representations for decision making aim to improve generalization and exploit additional sources of data. Recent work in the robotics domain have explored the use of pre-trained representations across multimodal data, including images and language, for downstream tasks (Karamcheti et al., 2023; Li et al., 2022; Ma et al., 2023a; Myers et al., 2023; Nair et al., 2022; Pari et al., 2022; Shah and Kumar, 2021; Cui et al., 2022; Jang et al., 2021). In reinforcement learning problems, representations are often trained to predict future states, rewards, goals, or actions (Anand et al., 2019; Ma et al., 2023b; Zhang et al., 2021; Fan et al., 2022), and can improve generalization and sample efficiency when used as value functions (Barreto et al., 2017; Blier et al., 2021; Dayan, 1993a; Dosovitskiy and Koltun, 2017; Choi et al., 2021). Some recent works have explored the use of additional structural constraints on representations to enable planning (Fang et al., 2022a; Zhang et al., 2022; Eysenbach et al., 2024; Hafner et al., 2019), or enforced metric properties to improve compositional generalization (Liu et al., 2023; Myers et al., 2024b; Wang et al., 2023).

The key distinction between our approach and past contrastive representation methods for robotics like VIP (Ma et al., 2023b), GRIF (Myers et al., 2023), and R3M (Nair et al., 2022) is that we focus on the real-world compositional generalization capabilities enabled by simply aligning representations across time in addition to the task modalities, without using the learned representations for policy extraction or defining a value function.

## 3 TEMPORAL REPRESENTATION ALIGNMENT

Given training on a series of short-horizon goal-reaching and instruction-following tasks, our goal is to learn a representation space such that our policy can generalize to a new (long-horizon) task that can be viewed as a sequence of known subtasks. We propose to structure this representation space by aligning the representations of states, goals, and language in a way that is more amenable to compositional generalization.

**Notation.** We take the setting of a goal- and language-conditioned MDP $\mathcal{M}$ with state space $\mathcal{S}$, continuous action space $\mathcal{A} \subseteq (0, 1)^{d_\mathcal{A}}$, initial state distribution $p_0$, dynamics $\mathrm{P}(s' \mid s, a)$, discount factor $\gamma$, and language task distribution $p_\ell$. A policy $\pi(a \mid s)$ maps states to a distribution over actions. We inductively define the $k$-step (action-conditioned) policy visitation distribution as:

$$p_1^\pi(s_1 \mid s_1, a_1) \triangleq p(s_1 \mid s_1, a_1),$$

$$p_{k+1}^\pi(s_{k+1} \mid s_1, a_1) \triangleq \int_\mathcal{A} \int_\mathcal{S} p(s_{k+1} \mid s, a) \, \mathrm{d}p_k^\pi(s \mid s_1, a_1) \, \mathrm{d}\pi(a \mid s)$$

$$p_{k+t}^\pi(s_{k+t} \mid s_t, a_t) \triangleq p^\pi(s_k \mid s_1, a_1). \tag{1}$$

Then, the discounted state visitation distribution can be defined as the distribution over $s^+$, the state reached after $K \sim \mathrm{Geom}(1 - \gamma)$ steps:

$$p_\gamma^\pi(s^+ \mid s, a) \triangleq \sum_{k=0}^\infty \gamma^k p_k^\pi(s^+ \mid s, a). \tag{2}$$

We assume access to a dataset of expert demonstrations $\mathcal{D} = \{\tau_i, \ell_i\}_{i=1}^K$, where each trajectory

$$\tau_i = \{s_{t,i}, a_{t,i}\}_{t=1}^H \in \mathcal{S} \times \mathcal{A} \tag{3}$$

is gathered by an expert policy $\pi^\mathrm{E}$, and is then annotated with $p_\ell(\ell_i \mid s_{1,i}, s_{H,i})$. Our aim is to learn a policy $\pi$ that can select actions conditioned on a new language instruction $\ell$. As in prior work (Walke et al., 2023), we handle the continuous action space by both our policy and the expert policy as an isotropic Gaussian with fixed variance; we will equivalently write $\pi(a \mid s, \varphi)$ or denote the mode as $\hat{a} = \pi(s, \varphi)$ for a task $\varphi$.

### 3.1 MOTIVATION: REPRESENTATIONS FOR REACHING DISTANT GOALS

We learn a goal-conditioned policy $\pi(a \mid s, g)$ that selects actions to reach a goal $g$ from expert demonstrations with behavioral cloning. Suppose we directly selected actions to imitate the expert on

two trajectories in $\mathcal{D}$:

$$
\left.
\begin{aligned}
s_1 \longrightarrow s_2 \longrightarrow \ldots \longrightarrow s_H \longrightarrow w \\[2em]
w \longrightarrow s'_1 \longrightarrow \ldots \longrightarrow s'_H \longrightarrow g
\end{aligned}
\right\} \quad \tau_i \in \mathcal{D} \tag{4}
$$

When conditioned with the composed goal $g$, we would be unable to imitate effectively as the composed state-goal $(s, g)$ is jointly out of the training distribution.

What *would* work for reaching $g$ is to first condition the policy on the intermediate waypoint $w$, then upon reaching $w$, condition on the goal $g$, as the state-goal pairs $(s_i, w)$, $(w, g)$, and $(s'_i, g)$ are all in the training distribution. If we condition the policy on some intermediate waypoint distribution $p(w)$ (or sufficient statistics thereof) that captures all of these cases, we can stitch together the expert behaviors to reach the goal $g$.

Consider the goal-conditioned behavioral cloning (Kaelbling, 1993) loss $\mathcal{L}_{\mathrm{BC}}^{\phi,\psi,\xi}$ conditioned with waypoints $w$.

$$
\mathcal{L}_{\mathrm{BC}}\big(\{s_i, a_i, s_i^+, g_i\}_{i=1}^K\big) = \sum_{i=1}^K \log \pi\big(a_i \mid s_i, \psi(g_i)\big). \tag{5}
$$

Enforcing the invariance needed to stitch Eq. (4) then reduces to aligning $\psi(g) \leftrightarrow \psi(w)$. The temporal alignment objective $\phi(s) \leftrightarrow \phi(s^+)$ accomplishes this indirectly by aligning both $\psi(w)$ and $\psi(g)$ to the shared waypoint representation $\phi(w)$:

$$
\mathcal{L}_{\mathrm{NCE}}\big(\{s_i, s_i^+\}_{i=1}^K; \phi, \psi\big) = \log\left(\frac{e^{\phi(s_i^+)^T \psi(s_i)}}{\sum_{j=1}^K e^{\phi(s_i^+)^T \psi(s_j)}}\right) + \sum_{j=1}^K \log\left(\frac{e^{\phi(s_i^+)^T \psi(s_i)}}{\sum_{i=1}^K e^{\phi(s_i^+)^T \psi(s_j)}}\right) \tag{6}
$$

## 3.2 Interfacing with Language Instructions

To extend the representations from Section 3.1 to compositional instruction following with language tasks, we need some way to ground language into the $\psi$ representation space. We use a similar approach to GRIF (Myers et al., 2023), which uses an additional CLIP-style (Radford et al., 2021) contrastive alignment loss with an additional pretrained language encoder $\xi$:

$$
\mathcal{L}_{\mathrm{NCE}}\big(\{g_i, \ell_i\}_{i=1}^K; \psi, \xi\big) = \sum_{i=1}^K \log\left(\frac{e^{\psi(g_i)^T \xi(\ell_i)}}{\sum_{j=1}^K e^{\psi(g_i)^T \xi(\ell_j)}}\right) + \sum_{j=1}^K \log\left(\frac{e^{\psi(g_i)^T \xi(\ell_i)}}{\sum_{i=1}^K e^{\psi(g_i)^T \xi(\ell_j)}}\right) \tag{7}
$$

## 3.3 Temporal Alignment

The Temporal Representation Alignment (TRA) approach structures the representation space of goals and language instructions to better enable compositional generalization. We learn encoders $\phi$, $\psi$, and $\xi$ to map states, goals, and language instructions to a shared representation space.

$$
\mathcal{L}_{\mathrm{NCE}}(\{x_i, y_i\}_{i=1}^K; f, h) = \sum_{i=1}^K \log\left(\frac{e^{f(y_i)^T h(x_i)}}{\sum_{j=1}^K e^{f(y_i)^T h(x_j)}}\right) + \sum_{j=1}^K \log\left(\frac{e^{f(y_i)^T h(x_i)}}{\sum_{i=1}^K e^{f(y_i)^T h(x_j)}}\right) \tag{8}
$$

$$
\mathcal{L}_{\mathrm{BC}}\big(\{s_i, a_i, s_i^+, \ell_i\}_{i=1}^K; \pi\big) = \sum_{i=1}^K \log \pi\big(a_i \mid s_i, \xi(\ell_i)\big) + \log \pi\big(a_i \mid s_i, \psi(s_i^+)\big) \tag{9}
$$

$$
\begin{aligned}
&\mathcal{L}_{\mathrm{TRA}}\big(\{s_i, a_i, s_i^+, g_i, \ell_i\}_{i=1}^K; \pi, \phi, \psi, \xi\big) \\
&= \underbrace{\mathcal{L}_{\mathrm{BC}}\big(\{s_i, a_i, s_i^+, \ell_i\}_{i=1}^K; \pi, \psi, \xi\big)}_{\text{behavioral cloning}} + \underbrace{\mathcal{L}_{\mathrm{NCE}}\big(\{s_i, s_i^+\}_{i=1}^K; \phi, \psi\big)}_{\text{temporal alignment}} + \underbrace{\mathcal{L}_{\mathrm{NCE}}\big(\{g_i, \ell_i\}_{i=1}^K; \psi, \xi\big)}_{\text{task alignment}}
\end{aligned} \tag{10}
$$

Note that the NCE alignment loss uses a CLIP-style symmetric contrastive objective (Radford et al., 2021; Eysenbach et al., 2024) — we highlight the indices in the NCE alignment loss (8) for clarity.

Our overall objective is to minimize Eq. (10) across states, actions, future states, goals, and language tasks within the training data:

$$\min_{\pi,\phi,\psi,\xi} \mathbb{E}_{\substack{(s_{1,i},a_{1,i},\ldots,s_{H,i},a_{H,i},\ell)\sim\mathcal{D} \\ i\sim\text{Unif}(1\ldots H) \\ k\sim\text{Geom}(1-\gamma)}} \left[ \mathcal{L}_{\text{TRA}}\left( \{s_{t,i}, a_{t,i}, s_{\min(t+k,H),i}, s_{H,i}, \ell\}_{i=1}^K; \pi, \phi, \psi, \xi \right) \right]. \quad (11)$$

---

**Algorithm 1:** Temporal Representation Alignment (TRA)

1: **input:** dataset $\mathcal{D} = (\{s_{t,i}, a_{t,i}\}_{t=1}^H, \ell_i)_{i=1}^N$
2: initialize networks $\Theta \triangleq (\pi, \phi, \psi, \xi)$
3: **while** training **do**
4:      sample a batch of transitions $\left\{ (s_{t,i}, a_{t,i}, s_{t+k,i}, \ell_i) \right\}_{i=1}^K \sim \mathcal{D}$ for $k \sim \text{Geom}(1-\gamma)$
5:      $\Theta \leftarrow (\pi, \phi, \psi, \xi) - \alpha \nabla_\Theta \mathcal{L}_{\text{TRA}}\left( \{s_{t,i}, a_{t,i}, s_{t+k,i}, \ell_i\}_{i=1}^K; \Theta \right)$
6: **output:** language $\ell$-conditioned policy $\pi(a_t|s_t, \xi(\ell))$
7:          goal $g$-conditioned policy $\pi(a_t|s_t, \psi(g))$

---

A summary of our approach is shown in Algorithm 1.

### 3.4 TEMPORAL ALIGNMENT AND COMPOSITIONALITY

We will formalize the intuition from Section 3.1 that TRA enables compositional generalization by considering the error on a "compositional" version of $\mathcal{D}$, denoted $\mathcal{D}^*$. Using the notation from Eq. (3), we can say $\mathcal{D}$ is distributed according to:

$$\mathcal{D} \triangleq \mathcal{D}^H \sim \prod_{i=1}^K p_0(s_{1,i}) p_\ell(\ell_i \mid s_{1,i}, s_{H,i}) \prod_{t=1}^H \pi^E(a_{t,i} \mid s_{t,i}) \, P(s_{t+1,i} \mid s_{t,i}, a_{t,i}), \quad (12)$$

or equivalently

$$\mathcal{D}^H \sim \prod_{i=1}^K p_0(s_{1,i}) p_\ell(\ell_i \mid s_{1,i}, s_{H,i}) \prod_{t=1}^H e^{\sigma^2 \|\pi^E(s_{t,i}) - a_{t,i}\|^2} \, P(s_{t+1,i} \mid s_{t,i}, a_{t,i}), \quad (13)$$

by the isotropic Gaussian assumption. We will define $\mathcal{D}^* \triangleq \mathcal{D}^{H'}$ to be a longer-horizon version of $\mathcal{D}$ extending the behaviors gathered under $\pi^E$ across a horizon $\alpha H \geq H' \geq H$ that additionally satisfies a "time-isotropy" property: the marginal distribution of the states is uniform across the horizon, i.e., $p_0(s_{1,i}) = p_0(s_{t,i})$ for all $t \in \{1 \ldots H'\}$.

We will relate the in-distribution imitation error $\text{ERR}(\bullet; \mathcal{D})$ to the compositional out-of-distribution imitation error $\text{ERR}(\bullet; \mathcal{D}^*)$. We define

$$\text{ERR}(\hat{\pi}; \tilde{\mathcal{D}}) = \mathbb{E}_{\tilde{\mathcal{D}}} \left[ \frac{1}{H} \sum_{t=1}^H \mathbb{E}_{\hat{\pi}} \left[ \|\tilde{a}_{t,i} - \hat{\pi}(\tilde{s}_{t,i}, \tilde{s}_{H,i})\|^2 / d_\mathcal{A} \right] \right] \quad (14)$$

$$\text{for} \quad \{\tilde{s}_{t,i}, \tilde{a}_{t,i}, \tilde{\ell}_i\}_{t=1}^H \sim \tilde{\mathcal{D}}. \quad (15)$$

On the training dataset this is equivalent to the expected behavioral cloning loss from Eq. (9).

**Assumption 1.** *The policy factorizes through inferred waypoints as:*

$$\textit{goals: } \pi(a \mid s, g) = \int \pi(a \mid s, w) \, P(s_t = w \mid s_{t+k} = g) \, dw \quad (16)$$

$$\textit{language: } \pi(a \mid s, \ell) = \int \pi(a \mid s, w) \, P(s_t = w \mid s_{t+k} = g) \, P(s_{t+k} = g \mid \ell) \, dw \, dg, \quad (17)$$

*where denote by $\pi(s, g)$ the MLE estimate of the action $a$.*

**Theorem 1.** *Suppose $\mathcal{D}$ is distributed according to Eq. (12) and $\mathcal{D}^*$ is distributed according to Eq. (12). When $\gamma > 1 - 1/H$ and $\alpha > 1$, for optimal features $\phi$ and $\psi$ under Eq. (11), we have*

$$\text{ERR}(\pi; \mathcal{D}^*) \leq \text{ERR}(\pi; \mathcal{D}) + \frac{\alpha - 1}{2\alpha} + \left( \frac{\alpha - 2}{2\alpha} \right) \mathbb{1}\{\alpha > 2\}. \quad (18)$$

We can also define a notion of the language-conditioned compositional generalization error:

$$\text{ERR}^{\ell}(\pi; \mathcal{D}^*) \triangleq \mathbb{E}_{\mathcal{D}^*}\left[\frac{1}{H}\sum_{t=1}^{H}\mathbb{E}_{\pi}\left[\|\tilde{a}_{t,i} - \pi(\tilde{s}_{t,i}, \tilde{\ell}_i)\|^2\right]\right]. \tag{19}$$

**Corollary 1.1.** *Under the same conditions as Theorem 1,*

$$\text{ERR}^{\ell}(\pi; \mathcal{D}^*) \leq \text{ERR}^{\ell}(\pi; \mathcal{D}) + \frac{\alpha - 1}{2\alpha} + \left(\frac{\alpha - 2}{2\alpha}\right)\mathbb{1}\{\alpha > 2\}. \tag{20}$$

The proofs as well as a visualization of the bound are in Appendix E.

## 4 EXPERIMENTS

Our experimental evaluation aims to answer the following research questions for TRA:

1. Can TRA enable zero-shot composition of multiple sequential tasks without additional prompting or planning methods?
2. How well does TRA perform compared to conventional offline RL algorithms in terms of task generalization and composition?
3. How well does TRA capture skills that are seen at a lower percentage within the dataset, compared to the numerous entries of object manipulation?
4. Is time alignment by itself sufficient for effective compositional generalization?

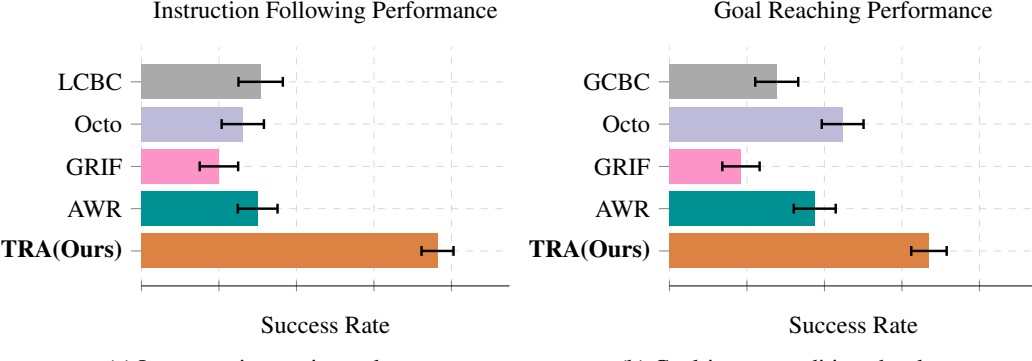

(a) Language instruction tasks      (b) Goal-image conditioned tasks

Figure 2: Aggregated performance on compositional generalization tasks, consisting of instruction-following and goal-reaching tasks.

Table 1: Compositional Generalization Error of Methods

| Modality | TRA | GRIF | LCBC | GCBC | Octo |
|---|---|---|---|---|---|
| image | $4.25 \pm 0.37$ | $5.24 \pm 0.34$ | — | $4.84 \pm 0.11$ | $5.15 \pm 0.41$ |
| language | $3.82 \pm 0.25$ | $4.95 \pm 0.32$ | $4.84 \pm 0.11$ | — | $4.56 \pm 0.32$ |

### 4.1 EXPERIMENTAL DETAILS

We evaluate TRA on a collection of held-out *compositionally-OOD* tasks – tasks for which the individual substeps are represented in the dataset, but the combination of those steps is unseen. For example, in a task such as "removing a bell pepper from a towel, and then sweep the towel", both the tasks "remove the bell pepper from the towel" and "sweep the towel" have similar entries within BridgeData, but such combined trajectory and language description does not exist. We utilize a real-world robot manipulation interface with a 7 DoF WidowX250 manipulator arm with 5Hz execution frequency. We train on an augmented version of the BridgeDataV2 dataset (Walke et al., 2023), which contains over 50k trajectories with 72k language annotations. We augment the dataset by rephrasing the language annotations, as described by (Myers et al., 2023), with 5 additional

rephrased language instruction for each language instruction present in the dataset, and randomly sample them during training.

In order to specifically test the ability of TRA to perform compositional generalization, we organize our evaluation tasks into 4 scenes that are unseen in BridgeData, each with increasing difficulty:

**Scene A – One-Step Drawer:** this is the only scene that are not compositionally-OOD, as all the tasks are one-step tasks. This scene involves opening, putting an item in, and closing a drawer. These tasks have been seen in BridgeData, although at a lower frequency than object manipulation, but the position in which they are initialized are unseen. They will be used to compare TRA's ability to baselines when solving single-step tasks.

**Scene B – Task Concatenation:** this scene involves concatenating multiple tasks of the same nature in sequence, where a robot must be able to perform all tasks within the same trajectory. During evaluation, we instruct the policy with instructions such as sweeping multiple objects in the scene that require composition (though are not sensitive to the *order* of the composition).

**Scene C – Semantic Generalization:** Unlike scene B, these tasks require manipulation with different objects of the same class. We test this using various food items seen within BridgeData and instruct the policy to put various food items within a container. An example of such task would be to have a table containing a banana, a sushi, a bowl, and various distractor objects, and instead of using specific language commands such as "put the banana and the sushi in the bowl," a more general statement such as "put the food items in a container" will be used.

**Scene D – Tasks with Dependency:** This is the most challenging of the set of tasks: these tasks have subtasks that require previous subtasks being completed for them to succeed. An example of this would be to open a drawer, and to take out an item in the drawer, as one cannot take out an item from the drawer if the drawer is not open.

The complete list of tasks is noted in Appendix C.

## 4.2 BASELINES

We compare against the following baselines:

**GRIF (Myers et al., 2023)** learns a goal- and language- conditioned policy using aligned goal image and language representations. In our experiments, this becomes equivalent to TRA when the temporal alignment objective is removed.

**GCBC (Walke et al., 2023)** learns a goal-conditioned behavioral cloning policy that concatenates the goal image with the image observation.

**LCBC (Walke et al., 2023)** learns a language-conditioned policy that concatenates the language with the image observation.

**OCTO (Ghosh et al., 2024)** uses a multimodal transformer to learn a goal- and language-conditioned policy. The policy is trained on Open-X dataset (O'Neill et al., 2024), which incorporates BridgeData in its entirety.

**AWR (Peng et al., 2019)** uses advantages produced by a value function to effectively extract a policy from an offline dataset. In this experiment, we use the difference between the contrastive loss between the current observation and the goal representation and the contrastive loss between the next observation and the goal representation as a surrogate for value function.

We train GRIF, GCBC, LCBC, and AWR using the same augmented Bridge Dataset as TRA, and we use an Octo-Base 1.5 model for our evaluation. A more detail approach is detailed in Appendix B. During evaluation, we give all policies the same goal state and language instruction regardless of the architecture, as they are trained on the same language instruction with the exception of Octo, which doesn't benefit from paraphrased language data, but does benefit from a more diverse language annotation set across a larger dataset of varying length and complexity.

## 4.3 EXPERIMENTAL EVALUATION

**Does TRA enable compositionality?** In Table 1, we compare the normalized mean squared error (MSE) of the TRA method with other methods on held-out compositionally-OOD image- and goal-specified tasks. These values are derived from passing the inputs through the policy network and sampling the mode of the distribution without unnormalizing the outputs based on the dataset. The

Table 2: Real-world Language Conditioned Evaluation

| | Task | **TRA** | GRIF | LCBC | Octo | AWR |
|---|---|---|---|---|---|---|
| | open the drawer | **0.80±0.1**[†] | 0.20±0.2 | **0.60±0.2** | **0.60±0.2** | 0.40±0.2 |
| | mushroom in drawer | **0.80±0.1** | **0.80±0.2** | 0.40±0.2 | 0.00±0.0 | **0.60±0.2** |
| | close drawer | **0.60±0.2** | **0.60±0.2** | **0.40±0.2** | **0.60±0.2** | **0.40±0.2** |
| (∗) | put the spoons on towels | **1.00±0.0** | 0.40±0.2 | 0.20±0.2 | 0.00±0.0 | 0.20±0.2 |
| (∗) | put the spoons on the plates | **0.80±0.2** | 0.20±0.2 | 0.20±0.2 | 0.20±0.2 | 0.00±0.0 |
| (∗) | fold cloth into the center | **1.00±0.0** | 0.20±0.2 | 0.40±0.2 | 0.40±0.2 | 0.40±0.2 |
| (∗) | sweep to the right | **0.80±0.1** | 0.20±0.2 | 0.40±0.2 | 0.40±0.2 | 0.00±0.0 |
| (∗) | put the corn and sushi on plate | **0.90±0.1** | 0.00±0.0 | 0.40±0.2 | 0.00±0.0 | 0.50±0.2 |
| (∗) | sushi and mushroom in bowl | **0.80±0.2** | 0.00±0.0 | **0.60±0.2** | 0.20±0.2 | **0.60±0.2** |
| (∗) | corn, banana, and sushi in bowl | **0.80±0.1** | 0.00±0.0 | 0.00±0.0 | 0.00±0.0 | 0.20±0.1 |
| (∗) | take the item out of the drawer | **0.60±0.2** | 0.00±0.0 | 0.00±0.0 | 0.20±0.2 | 0.00±0.0 |
| (∗) | move bell pepper and sweep towel | **0.50±0.2** | 0.00±0.0 | 0.00±0.0 | 0.20±0.2 | 0.00±0.0 |
| (∗) | corn on plate then sushi in pot | **0.70±0.1** | 0.00±0.0 | 0.40±0.2 | **0.60±0.2** | 0.20±0.2 |

[∗]indicates task is compositionally-OOD (has multiple steps never seen together in training)
[†]The best-performing method(s) up to statistical significance are **highlighted**

Table 3: Real-world Goal-Conditioned Evaluation

| | Task | **TRA** | GRIF | GCBC | Octo | AWR |
|---|---|---|---|---|---|---|
| | open the drawer | **0.60±0.2**[†] | 0.60±0.2 | **0.40±0.2** | **0.50±0.2** | **0.80±0.2** |
| | mushroom in drawer | **0.90±0.1** | 0.40±0.2 | **0.80±0.2** | **0.90±0.1** | **0.60±0.2** |
| | close drawer | **1.00±0.0** | 0.40±0.2 | 0.80±0.2 | 0.60±0.2 | 0.40±0.2 |
| (∗) | put the spoons on towels | **1.00±0.0** | 0.20±0.2 | 0.60±0.2 | 0.40±0.2 | 0.60±0.2 |
| (∗) | put the spoons on the plates | **1.00±0.0** | 0.00±0.0 | 0.40±0.2 | 0.00±0.0 | 0.80±0.2 |
| (∗) | fold cloth into the center | **1.00±0.0** | 0.00±0.0 | 0.00±0.0 | 0.60±0.2 | 0.00±0.0 |
| (∗) | sweep to the right | **0.70±0.1** | 0.40±0.2 | 0.00±0.0 | **0.80±0.2** | 0.00±0.0 |
| (∗) | put the corn and sushi on plate | **0.70±0.1** | 0.00±0.0 | 0.20±0.2 | 0.00±0.0 | 0.30±0.1 |
| (∗) | sushi and mushroom in bowl | **0.60±0.2** | 0.00±0.0 | 0.20±0.2 | **0.40±0.2** | **0.60±0.2** |
| (∗) | corn, banana, and sushi in bowl | **0.50±0.2** | 0.00±0.0 | 0.00±0.0 | **0.40±0.2** | 0.50±0.2 |
| (∗) | take the item out of the drawer | **0.40±0.2** | 0.00±0.0 | 0.00±0.0 | 0.20±0.2 | 0.00±0.0 |
| (∗) | move bell pepper and sweep towel | **0.60±0.2** | 0.20±0.2 | 0.20±0.2 | **0.40±0.2** | 0.00±0.0 |
| (∗) | corn on plate then sushi in pot | **0.30±0.1** | **0.20±0.2** | 0.00±0.0 | 0.00±0.0 | 0.00±0.0 |

[∗]indicates task is compositionally-OOD (has multiple steps never seen together in training)
[†]The best-performing method(s) up to statistical significance are **highlighted**

validation MSE for these tasks are lower with a statistically significant margin, demonstrating that in a compositionally-OOD setting, TRA provides a trajectory closer to expert demonstrations.

Section 4.2 and Section 4.2 show the success rates of the TRA method compared to other methods on real-world robot evaluation tasks. We marked all policies within the task orange if they achieve the best statistically significant performance. We first compare the performance against methods in Scene A. We observe that while TRA performs well with drawer tasks, its performance against baseline methods are not statistically significant. However, when being evaluated on compositionally-OOD **instruction following** tasks, TRA performs considerably better than that of any baseline methods.

While TRA completed 88.9% of tasks seen in Scene B, 83.3% of evaluations in Scene C, and 60% of tasks in Scene D with instruction following, the best-performing baseline for Scene B was 30% with

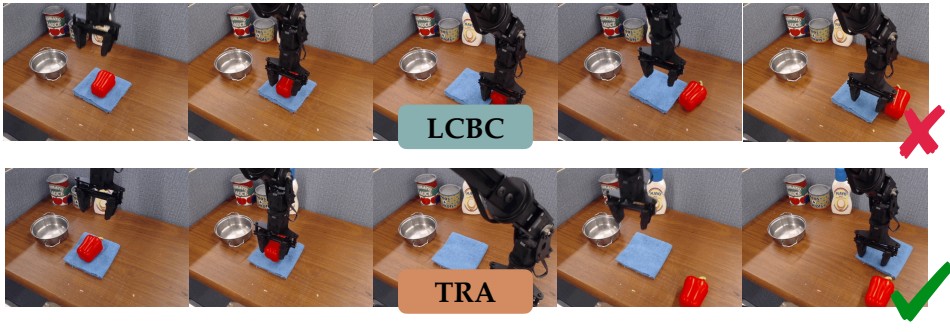

Figure 3: Example rollouts of a task with TRA and LCBC. While TRA is able to successfully compose the steps to complete the task, LCBC fails to ground the instruction correctly.

LCBC, 43.3% for Scene C with AWR, and 33.3% on Scene D with Octo. The same improvement was also present in goal reaching tasks, although at a lower level, in which Scene C produced 60% success rate and scene D produced a 43.3% success rate, as compared to 46.7% and 20% for the best-performing baselines.

Qualitatively, we see that policies trained under TRA provides a much smoother trajectory between different subtasks while following instructions, while other cannot replicate the same performance. Take removing the bell pepper + sweep task for example, with its visualization shown Fig. 3, while TRA was able to remove the bell pepper by grasping it and putting it to the bottom right corner of the table, LCBC cannot replicate the same performance, choosing to nudge the bell pepper instead and failed to execute the task.

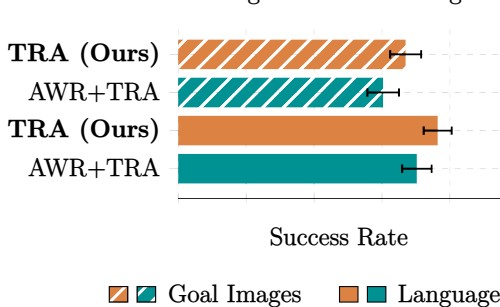

Figure 4: Aggregated success rate of using AWR as an additional policy learning metric over all 4 scenes.

**How well does TRA perform against Conventional Offline RL Algorithms?** While offline reinforcement learning promises good stitching behavior (Kumar et al., 2021), we demonstrate that TRA still outperforms offline reinforcement learning on robotic manipulation. Overall, TRA performs better than AWR for both language and image tasks, outperforming AWR by 45% on instruction following tasks, and by 25% on goal reaching tasks, showing considerable improvement over an offline RL method that promises compositional generalization via stitching.

Qualitatively, it is often seen that a policy trained with AWR would stop after one subtask, even though the goal instruction or image demanded all of the subtasks be completed. We can see this behavior in Fig. 1, in which we have the same goal image being fed in to 3 different policies in which all 3 food items must be put in the bowl. While TRA successfully completes all 3 subtasks, AWR chose to only complete one subtask and terminates right after putting the banana in the bowl. This is due to the fact that AWR on an offline dataset has a goal-reaching reward function, in which it does not attempt to align the representations of all trajectories across time unlike TRA.

**Does TRA help capturing rarely-seen skills within the dataset?** We also compare the performance of TRA against AWR across all scenes and compare the performance of the policies with all 3 tasks in Scene D as well as folding the towel, all rarely seen skills within BridgeData, as it mainly focused on object manipulation. When compared by task within language conditioned set, we discover AWR suffered a significant drop off in effectiveness, with its average success rate plummeting from 43.3% in Scene C compared to 6.67% in Scene D, while TRA had a smaller drop

off, from 83.3% to 60%, displaying that TRA generates better understanding of tasks that are rarely seen in the dataset. Other agents do not nearly achieve the same performance even as AWR in Scene D, as the lack of such compositional generalization prevented the policies from achieving all of the tasks at a reliable rate.

**Is TRA sufficient in achieving compositional generalization?** We demonstrate in our real-world experiment that only using temporal alignment is sufficient for achieving good compositional generalization. We evaluate this by comparing a policy trained on only temporal alignment loss (our method), and another policy trained on such loss and have these losses weighed by AWR.

Figure 4 shows that across all evaluation tasks, there exists no statistically significant difference between using and not using AWR in addition to temporal alignment, in fact, using AWR marginally decreases the efficacy of TRA, as compared to showing marginal improvement over vanilla GCBC methods and a similar performance with vanilla LCBC methods. While TRA qualitatively improve the smoothness of the execution trajectories, the same cannot be said about using AWR, in which after executing every subtask, the robot chose to return near the starting joint angles before executing the next subtask.

## 4.4 FAILURE CASES

While TRA provides an effective mechanism for compositional generalization, it is not immune to failures. Qualitatively, we observe that despite showing better compositional generalization, the policy still fails at a similar rate compared to other multivariate Gaussian policies when multimodal behavior is observed, other cases of early grasping and incorrect reaching are also observed at a similar rate. While TRA did provide marginal improvements as seen in Scene A, it does not provide full coverage of such scenarios. More analysis of failure cases can be seen in Appendix D.1.

## 5 CONCLUSIONS AND LIMITATIONS

In this paper, we studied the effects of adding a temporal representation alignment objective in behavior cloning, and we have discovered that by adding this metric, it allows a robot policy to perform robust compositional generalization even when the composition of such tasks are OOD.

**Limitations and Future Work** Due to restrictions placed by dataloaders, TRA cannot handle extremely long sequence of language, even though the difficulty of subtasks contained within the instructions may remain the same. Future work could also examine long-horizon tasks in bimanual or multi-agent settings, or investigate other properties like cross-embodiment generalization. To scale to these more complex settings, similar approaches with more complex architectures architectures such as transformers and diffusion policies may be needed for policy and/or representation learning. Future work could also examine combining TRA with hierarchical task decomposition using VLMs, or with other forms of structured task representations.

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

## A TRA IMPLEMENTATION

In this section, we provide some details on the implementatinon of temporal representation alignment (TRA) and its training process.

### A.1 DATASET CURATION

We use an augmented version of BridgeData. We augment the dataset by generating 5 additional paraphrased instruction per language instruction. During training process, we randomly sample the instructions for each trajectory to ensure an equal coverage of texts.

During data loading process, for each observation that is being sampled with timestep $k$, we also sample $k^+ \triangleq \min(k + x, H), x \sim \text{Geom}(1 - \gamma)$, and load $s_k$ along with $s_{k^+}$. We employ random cropping, resizing, and hue changes during training process image robustness.

### A.2 POLICY TRAINING

We use a ResNet-34 architecture for the policy network. We train our policy with one Google V4-8 TPU VM instance for 150,000 steps, which takes a total of 20 hours. We use a learning rate of $3 \times 10^{-4}$, 2000 linear warmup steps, and a MLP head of 3 layers of 256 dimensions after encoding the observation representations as well as goal representations.

## B    BASELINE IMPLEMENTATIONS

We summarize the implementation details of the baselines discussed in Section 4.2.

### B.1    OCTO

We use the Octo-base 1.5 model publicly available on HuggingFace for evaluating Octo baselines. We use inference code that is readily available for both image- and language- conditioned tasks. During inference, we use an action chunking window of 4 and an execution horizon window of 4.

### B.2    BEHAVIOR CLONING

We use the same architecture for LCBC as in Walke et al. (2023); Myers et al. (2023). During the training process we use the same hyperparameters as TRA.

### B.3    ADVANTAGE WEIGHTED REGRESSION

In order to train an AWR agent without separately implementing a reward critic, we follow Eysenbach et al. (2022) and use a surrogate for advantage:

$$\mathcal{A}(s_t) = \mathcal{L}_{\text{NCE}}\big(f(s_t), f(g)\big) - \mathcal{L}_{\text{NCE}}\big(f(s_{t+1}), f(g)\big). \tag{21}$$

Here, $f$ can be any of the encoders $\phi, \xi, \psi$. $\mathcal{L}$ is the same InfoNCE loss defined Section 3, and $g$ is defined as either the goal observation or the goal language instruction, depending on the modality.

And we extract the policy using advantage weighted regression (AWR) (Neumann and Peters, 2008):

$$\pi \leftarrow \arg\max_{\pi} \mathbb{E}_{s,a\sim\mathcal{D}}\Big[\log \pi(a|s, z) \exp\big(A(s, a)/\beta\big)\Big]. \tag{22}$$

During training, we set $\beta$ to 1, and we use a batch size of 128, the same value as policy training for our method.

## C    EXPERIMENT DETAILS

In this section, we go through our experiment details and how they are set up. During evaluation, we randomly reset the positions of each item within the table, and perform 5 to 10 trials on each task, depending on whether this task is important within each scene. We examine tasks that are seen in BridgeData, which include conventionally less challenging tasks such as object manipulation, and challenging tasks to learn within the dataset such as cloth folding and drawer opening.

### C.1    LIST OF TASKS

Table 4 describes each task within each scene, and the language annotation used when the policy is used for inference. Every task that is outside of the drawer scene are multiple step, and require compositional generalization.

### C.2    INFERENCE DETAILS

During inference, we use a maximum of 200 timesteps to account for long-horizon behaviors, which remains the same for all policies. We determine a task as successful when the robot completes the task it was instructed to within the timeframe. For evaluating baselines, we use 5 trials for each of the tasks.

### C.3    VALIDATION MSE

In addition to rolling out the policy on real-world robot settings, we additionally collected 9 additional tasks that are compositionally OOD for 5 trajectories each, and we use 3 randomly selected seeds to train policies to evaluate the MSE on the validation trajectories.

Table 4: Task Instructions

| Scene | Count | Task Description | Instruction |
|---|---|---|---|
| Drawer | 10 | open the drawer | "open the drawer" |
| | 10 | put the mushroom in the drawer | "put the mushroom in the drawer" |
| | 10 | close the drawer | "close the drawer" |
| Task Generalization | 5 | put the spoons on the plates | "move the spoons onto the plates." |
| | 5 | put the spoons on the towels | "move the spoons on the towels" |
| | 6 | fold the cloth into the center from all corners | "fold the cloth into center" |
| | 10 | sweep the towels to the right | "sweep the towels to the right of the table" |
| Semantic Generalization | 10 | put the sushi and the corn on the plate | "put the food items on the plate" |
| | 5 | put the sushi and the mushroom in the bowl | "put the food items in the bowl" |
| | 10 | put the sushi, corn, and the banana in the bowl | "put everything in the bowl" |
| Tasks With Dependency | 10 | take mushroom out of drawer | "open the drawer and then take the mushroom out of the drawer" |
| | 10 | move bell pepper and sweep towel | "move the bell pepper to the bottom right corner of the table, and then sweep the towel to the top right corner of the table" |
| | 10 | put the corn on the plate, *and then* put the sushi in the pot | "put the corn on the plate and then put the sushi in the pot" |

"fold the towel into center"

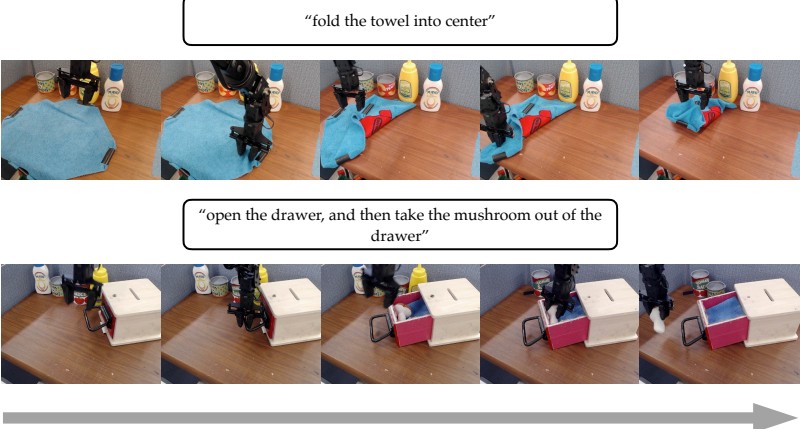

"open the drawer, and then take the mushroom out of the drawer"

"put the corn on the plate, and then put the sushi in the pot"

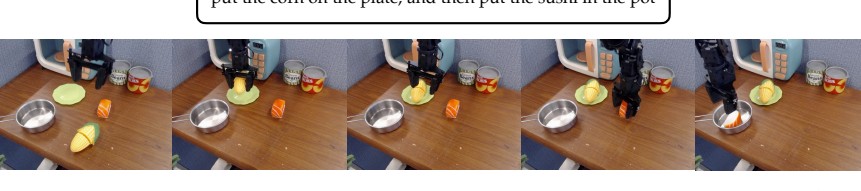

Figure 5: In these figures, we see that TRA is able to perform good compositional generatlization over a variety of tasks seen within BridgeData

## D  ADDITIONAL VISUALIZATIONS

In this section, we show additional visualizations of TRA's execution on compositionally-OOD tasks. We use *folding, taking mushroom out of the drawer*, and *corn on plate, then sushi in the pot* as examples, as these tasks require a strong degree of dependency to complete at Appendix D.

### D.1  FAILURE CASES

We break down failure cases in this section. While TRA performs well in compositional generalization, it cannot counteract against previous failures seen with behavior cloning with a Gaussian Policy.

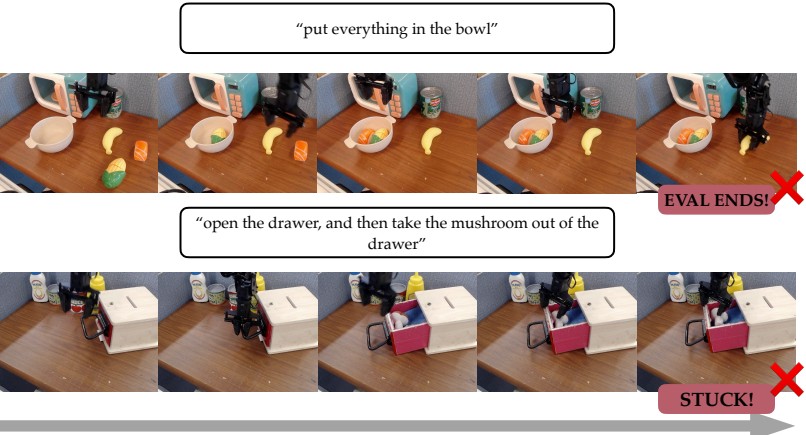

Figure 6: Most of the failure cases came from the fact that a policy cannot learn depth reasoning, causing early grasping or late release, and it has trouble reconciling with multimodal behavior

# E   ANALYSIS OF COMPOSITIONALITY

We prove the results from Section 3.4.

## E.1   GOAL CONDITIONED ANALYSIS

**Theorem 1.** *Suppose $\mathcal{D}$ is distributed according to Eq. (12) and $\mathcal{D}^*$ is distributed according to Eq. (12). When $\gamma > 1 - 1/H$ and $\alpha > 1$, for optimal features $\phi$ and $\psi$ under Eq. (11), we have*

$$\text{ERR}(\pi; \mathcal{D}^*) \le \text{ERR}(\pi; \mathcal{D}) + \frac{\alpha - 1}{2\alpha} + \left(\frac{\alpha - 2}{2\alpha}\right)\mathbb{1}\{\alpha > 2\}. \tag{18}$$

*Proof.* We have from Eq. (15) for $K \sim \text{Geom}(1 - \gamma)$:

$$\text{ERR}(\pi; \mathcal{D}^*) \triangleq \mathbb{E}_{\mathcal{D}^*}\left[\frac{1}{H'}\sum_{t=1}^{H'}\frac{\|\tilde{a}_{t,i} - \pi(\tilde{s}_{t,i}, \tilde{g}_i)\|^2}{n_{d_{\mathcal{A}}}}\right]$$

$$= \frac{1}{H'}\mathbb{E}_{\mathcal{D}^*}\left[\sum_{t=1}^{H'-2H}\frac{\|\tilde{a}_{t,i} - \pi(\tilde{s}_{t,i}, \tilde{g}_i)\|^2}{n_{d_{\mathcal{A}}}}\right] + \frac{1}{H'}\mathbb{E}_{\mathcal{D}^*}\left[\sum_{H'-2H+1}^{H'-H}\frac{\|\tilde{a}_{t,i} - \pi(\tilde{s}_{t,i}, \tilde{g}_i)\|^2}{n_{d_{\mathcal{A}}}}\right]$$

$$+ \frac{1}{H'}\mathbb{E}_{\mathcal{D}^*}\left[\sum_{t=H'-H+1}^{H'}\frac{\|\tilde{a}_{t,i} - \pi(\tilde{s}_{t,i}, \tilde{g}_i)\|^2}{n_{d_{\mathcal{A}}}}\right]$$

$$\le \frac{1}{H'}\mathbb{E}_{\mathcal{D}^*}\left[\sum_{t=H'-H+1}^{H'}\frac{\|\tilde{a}_{t,i} - \pi(\tilde{s}_{t,i}, \tilde{g}_i)\|^2}{n_{d_{\mathcal{A}}}}\right] + \frac{1}{H'}\mathbb{E}_{\mathcal{D}^*}\left[\sum_{t=H'-2H+1}^{H'-H}\frac{\|\tilde{a}_{t,i} - \pi(\tilde{s}_{t,i}, \tilde{g}_i)\|^2}{n_{d_{\mathcal{A}}}}\right]$$

$$+ \left(\frac{\alpha - 2}{2\alpha}\right)\mathbb{1}\{\alpha > 2\}$$

$$\le \frac{1}{H'}\mathbb{E}_{\mathcal{D}^*}\left[\sum_{t=H'-H+1}^{H'}\frac{\|\tilde{a}_{t,i} - \pi(\tilde{s}_{t,i}, \tilde{s}_{H',i})\|^2}{n_{d_{\mathcal{A}}}}\right]$$

$$+ \frac{1}{H'}\mathbb{E}_{\mathcal{D}^*}\left[\sum_{t=H'-2H+1}^{H'-H}\mathbb{E}_K\left[\frac{\|\tilde{a}_{t,i} - p^\pi(\tilde{s}_{t,i}|\tilde{s}_{H'-K,i})\|^2}{n_{d_{\mathcal{A}}}}\right]\right] + \left(\frac{\alpha - 2}{2\alpha}\right)\mathbb{1}\{\alpha > 2\}$$

$$\leq \frac{1}{H'} \, \mathbb{E}_{\mathcal{D}^*} \Big[ \sum_{t=H'-H+1}^{H'} \frac{\|\tilde{a}_{t,i} - \pi(\tilde{s}_{t,i}, \tilde{s}_{H',i})\|^2}{n_{d_{\mathcal{A}}}} \Big]$$

$$+ \frac{1}{H'} \, \mathbb{E}_{\mathcal{D}^*} \Big[ \sum_{t=H'-2H+1}^{H'-H} \mathbb{E}_K \Big[ \frac{\|\tilde{a}_{t,i} - p^\pi(\tilde{s}_{t,i} | \tilde{s}_{H'-K,i})\|^2}{n_{d_{\mathcal{A}}}} \Big] \Big] + \Big( \frac{\alpha - 2}{2\alpha} \Big) \mathbb{1}\{\alpha > 2\}$$

$$\leq \frac{1}{H'} \, \mathbb{E}_{\mathcal{D}^*} \Big[ \sum_{t=H'-H+1}^{H'} \frac{\|\tilde{a}_{t,i} - \pi(\tilde{s}_{t,i}, \tilde{s}_{H',i})\|^2}{n_{d_{\mathcal{A}}}} \Big]$$

$$+ \frac{1}{H'} \, \mathbb{E}_{\mathcal{D}^*} \Big[ \sum_{t=H'-2H+1}^{H'-H} \mathbb{E}_K \Big[ \frac{\|\tilde{a}_{t,i} - p^\pi(\tilde{s}_{t,i} | \psi(\tilde{s}_{H'-K,i}))\|^2}{n_{d_{\mathcal{A}}}} \Big] \Big] + \Big( \frac{\alpha - 2}{2\alpha} \Big) \mathbb{1}\{\alpha > 2\}$$

$$\leq \mathrm{ERR}(\pi; \mathcal{D}^*) + \frac{1}{H'} \, \mathbb{E}_{\mathcal{D}^*} \Big[ \frac{1 - \gamma^H}{1 - \gamma} \Big] + \Big( \frac{\alpha - 2}{2\alpha} \Big) \mathbb{1}\{\alpha > 2\}$$

$$\leq \mathrm{ERR}(\pi; \mathcal{D}^*) + \frac{\alpha - 1}{2\alpha} + \Big( \frac{\alpha - 2}{2\alpha} \Big) \mathbb{1}\{\alpha > 2\}. \tag{23}$$

$$\square$$

## E.2 LANGUAGE CONDITIONED ANALYSIS

**Corollary 1.1.** *Under the same conditions as Theorem 1,*

$$\mathrm{ERR}^\ell(\pi; \mathcal{D}^*) \leq \mathrm{ERR}^\ell(\pi; \mathcal{D}) + \frac{\alpha - 1}{2\alpha} + \Big( \frac{\alpha - 2}{2\alpha} \Big) \mathbb{1}\{\alpha > 2\}. \tag{20}$$

The proof is similar to Appendix E.1, but over the predictions of $\xi$ instead of $\psi$.

## E.3 VISUALIZING THE BOUND

We compare the bound from Theorem 1 with the "worst-case" bound of $\mathrm{ERR}(\pi; \mathcal{D}^*) - \mathrm{ERR}(\pi; \mathcal{D})$ in Appendix E.3. The bound from Theorem 1 is tighter than the worst-case bound, and it shows that the compositional generalization error decreases as $\alpha$ increases.

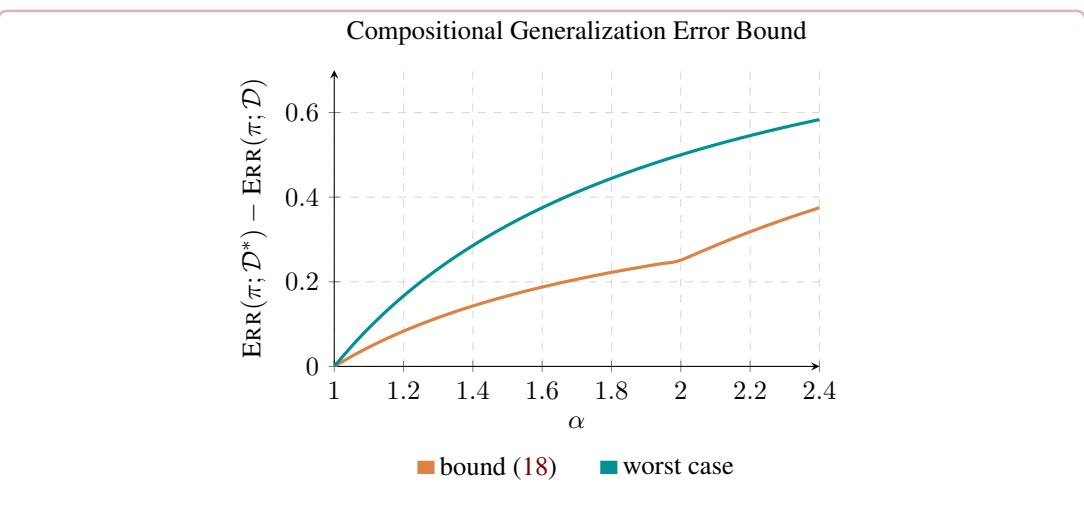

Figure 7: Visualizing the bound (Eq. 18 from Theorem 1) on the compositional generalization error.

