# OpenReview forum: "Successor Representations Enable Emergent Compositional Instruction Following"
_ICLR.cc/2025/Conference — Submitted to ICLR 2025_

### Official Review · Reviewer_xptZ · 2024-10-31

**Soundness:** 3
**Presentation:** 3
**Contribution:** 3
**Rating:** 6
**Confidence:** 3

**Summary:**

This paper introduces Temporal Representation Alignment (TRA), a method for enabling compositional generalization in robotic manipulation tasks without explicit subtask planning or reinforcement learning. The key idea is to add a temporal alignment loss that encourages the policy to learn structured representations that capture temporal relationships between states. The authors evaluate TRA on a range of tabletop manipulation tasks using the BridgeData setup, showing improved performance on compositionally novel tasks specified through either language instructions or goal images. The main contribution is demonstrating that adding this auxiliary temporal alignment objective during training can enable a policy to implicitly decompose and execute multi-step tasks, even when the specific sequence of steps was never seen during training.

**Strengths:**

1） Novel approach that achieves compositional generalization without requiring explicit hierarchical structure or planning, demonstrating that temporal alignment of representations is sufficient
2） Comprehensive empirical evaluation across multiple task types and comparison against strong baselines
3） Clear ablation studies that validate the importance of the temporal alignment component

**Weaknesses:**

## Comparative Analysis:

- Need stronger justification for why TRA is preferable to VLM/LLM-based decomposition approaches


## Evaluation Limitations


- Tasks are limited to relatively simple manipulation scenarios in a highly controlled environment

- Missing comparison with recent language model-based task decomposition methods (e.g., RT-H)


## Methodological Comparison Gaps:

- The paper's main contribution focuses on compositional long-horizon tasks, but doesn't adequately compare against state-of-the-art VLM/LLM-based task decomposition methods
- No clear demonstration of advantages over approaches that use large language models for task decomposition combined with foundation models like Octo or OpenVLA for sub-task execution
- Missing analysis of computational efficiency compared to VLM/LLM-based approaches


## Insufficient Analysis:

- Limited theoretical analysis of when/why temporal alignment enables compositional generalization

**Questions:**

1. How sensitive is the method to the choice of discount factor γ in the temporal alignment objective? Was any ablation done on this hyperparameter?



2. For the semantic generalization experiments (Scene C), how robust is the method to variations in object appearance beyond what was seen in training?

3. How does TRA compare to recent LLM-based task decomposition methods (like RT-H) in terms of:




4. Could you provide quantitative comparisons with methods that use LLMs for task decomposition combined with foundation models (like Octo/OpenVLA) for execution?

5. What advantages does TRA offer over LLM-based decomposition approaches for long-horizon tasks? Please provide concrete examples and experimental results.

6How does the method scale to more complex real-world scenarios with greater environmental variation and uncertainty?

---

> ### Author Response · Authors · 2024-11-27
> **Rebuttal**
>
> Thank you for your detailed response and thoughtful feedback. We would like to address your remaining concerns below. **Please inform us if you have any remaining concerns.**
>
> > Limited theoretical analysis of when/why temporal alignment enables compositional generalization
>
> **We have added theoretical justification for why temporal alignment enables task compositionality** to section 3.4, ''Temporal alignment and Compositionality.'' Intuitively, if goal representations are aligned with the previous states that lead to them, then goals which are slightly beyond the horizon of in-distribution tasks have a high probability of being in in-distribution (assuming the contrastive loss objective is minimized; see the section and the appendix for the full formalism). The key result is [Theorem 1](https://gcdnb.pbrd.co/images/ymsaYvcUTr8Z.png), which bounds the compositional out-of-distribution error of the policy in terms of the in-distribution error based on $\alpha=H'/H$, or how much larger the horizon $H'$ of the composed tasks is relative to $H$, the training horizon. We check this bound is meaningful by plotting it against a naive ''worst-case'' compositional generalization bound, showing it is meaningfully tighter in the [new Figure 7](https://gcdnb.pbrd.co/images/o0IVc7cvssOp.png). The result can be extended to compositionally-OOD language in addition to goals ([Corollary 1](https://gcdnb.pbrd.co/images/cHBZRXhhxjZy.png)).
>
> > How sensitive is the method to the choice of discount factor $\gamma$ in the temporal alignment objective? Was any ablation done on this hyperparameter?
>
> In our preliminary experiments, we found there was not too much sensitivity to the precise value of $\gamma$, as long as $\frac{1}{1-\gamma}$ was proportional to the horizon $H$. This is supported by the [new theoretical results](https://gcdnb.pbrd.co/images/ymsaYvcUTr8Z.png) which require $\frac{1}{1-\gamma}>H$ for the benefits in compositional generalization.
>
> > For the semantic generalization experiments (Scene C), how robust is the method to variations in object appearance beyond what was seen in training?
>
> Our contribution is to study how structured representation learning objectives can enable *compositional generalization*. The related notion of *object-level generalization* is a distinct property, depending on the model architecture (ResNet), the diversity of the dataset (Bridge v2), and the generalization capabilities of the pre-trained visual embeddings (CLIP). We observe some robustness over object appearance, in line with the observations of past work in the Bridge v2 setting that has focused on other forms of generalization (e.g., SuSIE [1], GRIF [2], RT-X [3]).
>
> > How does the method scale to more complex real-world scenarios with greater environmental variation and uncertainty?
>
> We have added discussion to our ''Limitations and future work'' section on how future work could explore similar compositionality properties in more complex environments such as bimanual, multi-agent, or cross-embodiment settings. Note that the tasks explored in this paper are of comparable difficulty to past works published at ICLR that use the BridgeDataset (v2) setting [1,4].

---

> ### Author Response · Authors · 2024-11-27
> **Rebuttal (cont.)**
>
> > Need stronger justification for why TRA is preferable to VLM/LLM-based decomposition approaches
>
> > Could you provide quantitative comparisons with methods that use LLMs for task decomposition combined with foundation models (like Octo/OpenVLA) for execution?
>
> > What advantages does TRA offer over LLM-based decomposition approaches for long-horizon tasks? Please provide concrete examples and experimental results.
>
> There have been many works in recent years that have used pipelines incorporating VLM/LLMs with low-level control policies to enable long-horizon behaviors [5,6,7,8,9]. The advantage of end-to-end approaches like TRA is they avoid compounding errors from multi-stage pipelines, and do not depend on large pre-trained models that need to be fine-tuned and prompted correctly. Note that representation-learning approaches like TRA still have access to some of general knowledge from VLM training by using pre-trained language and vision representations, but do not suffer the costs of hierarchical decomposition in doing so.
>
> Since our focus in this paper is how the structure of successor representations can enable improved end-to-end compositional reasoning, we restricted our comparisons to be across end-to-end approaches. For quantitative results with VLM decomposition, we would like to refer to the evaluations in [PALO](https://arxiv.org/pdf/2408.16228v1) [7], which feature some similar long-horizon tasks within the Bridge v2 dataset. While PALO achieves impressive long-horizon performance, it requires significant prompt engineering ([7], Appendix F) and expert demonstrations of all the new tasks. Similar limitations are seen with methods like RT-H [9], Inner Monologue [6], and SayCan [5].
>
> In theory, VLM/LLM planners could be combined with TRA to solve even longer tasks when they are available. We have added discussion to the ''Limitations and related work'' section on these directions. **Does this resolve your concern? Please let us know if there are any further comparisons we could provide.**
>
> ### References
>
> [1] Black, K. et al., 2024. ''Zero-Shot Robotic Manipulation With Pre-Trained Image-Editing Diffusion Models.'' *ICLR*
>
> [2] Myers, V. et al., 2023. ''Goal Representations for Instruction Following: A Semi-Supervised Language Interface to Control.'' *CoRL*
>
> [3] O'Neill, A. et al., 2024. ''Open X-Embodiment: Robotic Learning Datasets and RT-X Models.'' *ICRA*
>
> [4] Zheng, C. et al., 2024. ''Stabilizing Contrastive RL: Techniques for Robotic Goal Reaching From Offline Data.'' *ICLR*
>
> [5] Ahn, M. et al., 2022. ''Do as I Can, Not as I Say: Grounding Language in Robotic Affordances.'' *CoRL*
>
> [6] Huang, W. et al., 2022. ''Inner Monologue: Embodied Reasoning Through Planning With Language Models.'' *CoRL*
>
> [7] Myers, V. et al., 2024. ''Policy Adaptation via Language Optimization: Decomposing Tasks for Few-Shot Imitation.'' *CoRL*
>
> [8] Zawalski, M. et al., 2024. ''Robotic Control via Embodied Chain-of-Thought Reasoning.'' *CoRL*
>
> [9] Belkhale, S. et al., 2024. ''RT-H: Action Hierarchies Using Language.'' arXiv:2403.01823

---

> > ### Comment · Reviewer_xptZ · 2024-11-28
> > **Thanks for the reply**
> >
> > Thanks for the reply, your answer solved most of my queries and given that I am already at POSITIVE score at the moment, I will keep my score the same. I would still recommend doing more fly pick and place tasks, the current experimental results do not highlight its advantages over the decomposed scheme

---

> ### Comment · Area_Chair_9yuw · 2024-11-27
> **Rebuttal Response**
>
> Dear Reviewer,
> Do you mind letting the authors know if their rebuttal has addressed your concerns and questions? Thanks!
> -AC

---

### Official Review · Reviewer_4vMh · 2024-11-04

**Soundness:** 2
**Presentation:** 3
**Contribution:** 2
**Rating:** 5
**Confidence:** 4

**Summary:**

The paper introduces a self-supervised loss aimed at improving compositionality in language- and goal-image conditioned robot policies. The approach leverages contrastive learning with the NCE objective between states of similar trajectories while simultaneously aligning goal embeddings from language and image inputs. This improves compositional generalization and is tested in 4 experiment settings in the real world.

**Strengths:**

- simple extension to improve learning of policies using SSL

- strong results in real robot experiments

- easy to use for existing policy frameworks

**Weaknesses:**

Weaknesses:
- Technical omissions:
    - No theoretical foundation for why temporal alignment should enable task compositionality is provided. Experiments on a single dataset do not provide enough empirical evidence to justify the claims of the paper.
    - Given the weak theoretical justification of the made claims experiments on a single dataset are not enough to verify those. More experiments in reproducible benchmarks are necessary or detailed theoretical discussion why aligning similar states should result in this compositional generalization.
- Limited experimental validation:
    - Evaluation restricted to a single real world kitchen dataset
    - No testing in reproducible benchmark environments (e.g., CALVIN, SIMPLER or similar simulators) that would enable fair comparison with future work. Given the huge performance gain of TRA compared to GRIF and other baselines I am interested to see how it performs in other domains.
    - Table 1 does not provide evidence for the made claims as just predicting trajectories without actual rollouts can be very misleading and robotics and does not have a clear correlation with success rate.
- Writing and clarity issues:
    - Incomplete sentence in Chapter 3 disrupts flow
    - Complex, run-on sentences throughout make technical content difficult to understand
    - Overall writing requires substantial revision for clarity and coherence
    - Loss function lacks clear explanation and intuition

Summary:
The paper's potential contributions are undermined by unclear writing, missing technical details, and limited experimental evaluation. The paper does not provide any theoretical justification on the gains of the method. Since experiments are limited to a single non-reproducible real world benchmark, there is not a lot of empirical evidence to support these claims. While I acknowledge the number of real world experiments and the related effort to test these, they are still coming from a single dataset. Given the big performance gains shown (+60% compared to second best baselines in a setting), I expect to see similar results in other simulation domains.
Major revisions needed to address the following issues:
1. Improve writing clarity and technical explanations
2. Expand experimental validation across multiple environments
3. Include theoretical justification for the proposed claims

**Questions:**

- Can you test the proposed method on established in reproducible simulation benchmarks like CALVIN and SIMPLER to provide more empirical evidence for the claims of the paper?

- How big is the computation overhead for the proposed method?

- Can you provide some theoretical analysis to why the proposed ssl loss enables compositional generalization by aligning similar states?

- Performance of GRIF reported in the original paper for the same task is very different compared to the reported values here: "put the spoons on towels" from GRIF 0.9 and here "put the spoon on the towel" 0.2. How do you explain these big gaps?

---

> ### Author Response · Authors · 2024-11-24
> **Rebuttal**
>
> Thank you for your detailed and thoughtful review.
>
> It seems your primary concerns relate to (1) a lack of theoretical justification for the method, (2) insufficient experimental evidence, and (3) issues with presentation. Please let us know if these changes address your concerns.
> - For (1): **we have added a new theory section (3.4),** where we show how the TRA objective can induce a tighter bound on compositional generalization under some assumptions ([see new Theorem 1](https://gcdnb.pbrd.co/images/ymsaYvcUTr8Z.png))
> - For (2): **we have run additional ablations to show the robustness of TRA at real-world compositional generalization.**
> - For (3): we have made general improvements to writing with key **changes highlighted in red** based on your feedback.
>
> > No theoretical foundation for why temporal alignment should enable task compositionality is provided. Experiments on a single dataset do not provide enough empirical evidence to justify the claims of the paper without theoretical results.
>
> **We have added theoretical justification for why temporal alignment enables task compositionality** to section **3.4**, "Temporal alignment and Compositionality."
> Intuitively, if goal representations are aligned with the previous states that lead to them, then goals which are slightly beyond the horizon of in-distribution tasks have a high probability of being in in-distribution (assuming the contrastive loss objective is minimized; see the section and the appendix for the full formalism).
> The key result is [Theorem 1](https://gcdnb.pbrd.co/images/ymsaYvcUTr8Z.png), which bounds the compositional out-of-distribution error of the policy in terms of the in-distribution error based on $\alpha=H'/H$, or how much larger the horizon $H'$ of the composed tasks is relative to $H$, the training horizon.
> We check this bound is meaningful by plotting it against a naive "worst-case" compositional generalization bound, showing it is meaningfully tighter in the [new Figure 7](https://gcdnb.pbrd.co/images/o0IVc7cvssOp.png).
> The result can be extended to compositionally-OOD language in addition to goals ([Corollary 1](https://gcdnb.pbrd.co/images/cHBZRXhhxjZy.png)).
> Do these theoretical claims address your concern?
>
> > Loss function lacks clear explanation and intuition
>
> We have added a section, "**3.1** Motivation: Representations for Reaching Distant Goals," to provide intuition and explanation for the loss function. This intuition is then formalized in "**3.4** Temporal alignment and Compositionality," where we show how using these representations can tighten the bound on the compositional generalization error under some assumptions ([see new Theorem 1](https://gcdnb.pbrd.co/images/ymsaYvcUTr8Z.png)).
>
> > Additional experiments
>
> We have run some additional ablations to show the robustness of TRA for real-world compositional generalization ([see Table here](https://gcdnb.pbrd.co/images/UfZ0kvWXhaxd.png))
>
> > How big is the computation overhead for the proposed method?
>
> The computational overhead from TRA is relatively small ($<25\%$). The main contributor to this is the temporal alignment loss, since it requires forward and backward passes through the additional learned $\phi$ representation.
>
> > Performance of GRIF reported in the original paper for the same task is very different compared to the reported values here: "put the spoons on towels" from GRIF 0.9 and here "put the spoon on the towel" 0.2. How do you explain these big gaps?
>
> Note the difference in the two tasks: "put the spoon**s** on towels" (plural) v.s. "put the spoon on the towel." The plural version of the task requires compositional generalization as there are no examples of manipulating multiple spoons at the same time in the dataset. **The large gap is precisely because GRIF does not enable better compositional generalization, unlike TRA.**
>
> > Writing and clarity issues:
>
> Thank you for noting these issues.
> We have substantially revised the text based on your comments and made general improvements to writing. Please let us know if any of the writing is still unclear.
>
> > Table 1 does not provide evidence for the made claims as just predicting trajectories without actual rollouts can be very misleading and robotics and does not have a clear correlation with success rate.
>
> Yes, our main result is the substantial improvement in actual rollouts shown in Figure 2 and Tables 2 and 3. We included Table 1 to provide additional quantitative evidence for compositional generalization (in line with the definitions of generalization error in the new Theorem 1). If you would prefer, we could also cut Table 1 entirely or move it to the appendix.
> **Would this address your concern?**

---

> > ### Comment · Reviewer_4vMh · 2024-11-25
> >
> > Thank you for your effort in the rebuttal. The updates in the Main Method Section do improve the paper a lot and provide needed clarification and motivation.
> >
> > The request for adding a simulation based results on CALVIN or similar benchmark for reproducibility and future work still remain open.
> >
> > Could you also elaborate in more detail the theoretical novelties and differences of TRA compared to similar prior work like VIP, LIV or Inference via Interpolation that also employ contrastive representation learning in the policy latent space? How does your loss formulation differ from these prior work?
> >
> > Given the changes to the main part of the paper and the open remaining concerns, I increase my score to 5 for now.

---

> ### Author Response · Authors · 2024-12-04
> **Additional Response**
>
> Thank you for the response!
>
> > The request for adding a simulation based results on CALVIN or similar benchmark for reproducibility and future work still remain open.
>
> Thank you for this suggestion. We believe that simulation results in a benchmark that explicitly tests compositional behaviors would be a valuable addition. In practice, many standard (offline) RL benchmarks actually fail to adequately test this property [1].
>
> Since the time of our initial submission, a new offline RL benchmark, OGBench [2], has been released that explicitly tests compositionality and contains several manipulation tasks and datasets. We have implemented the TRA approach in this environment, and plan to include the full comparisons in a future revision.
>
> > Could you also elaborate in more detail the theoretical novelties and differences of TRA compared to similar prior work like VIP, LIV or Inference via Interpolation that also employ contrastive representation learning in the policy latent space? How does your loss formulation differ from these prior work?
>
> There are several key differences between TRA and past representation learning approaches for robotics. At a high-level, TRA focuses on the question of *compositionality*. We show that a simple and scalable time-contrastive representation learning objective can produce this property when used for *task representation*, without requiring any explicit planning or reinforcement learning. In contrast, prior approaches like VIP and Inference via Interpolation focus on learning state representations to improve other aspects of decision making. We briefly discuss some of the technical distinctions relative to the mentioned approaches below.
>
> **VIP**:
>
> VIP [3] focuses on learning representations such that distances in the representation space correspond to goal-conditioned value functions. This work makes several assumptions about the environment: (1) that dynamics are fully deterministic, and (2) that goal-conditioned values are symmetric (i.e., $V(s;g)=V(g;s)$). Note that both of these assumptions are violated in real-settings: (1) real world dynamics are noisy/unpredictable, and (2) any external force acting on the system (e.g., gravity) breaks symmetry of goal-conditioned value functions—this is most dramatic with actions like opening the gripper, which at an elevation can drop an object, which is much easier to do than undo.
>
> Under these assumptions, the VIP objective is a Fenchel dual of a time-contrastive representation objective. TRA does not require either of these assumptions, which may help with long-horizon tasks that are compositionally OOD and require awareness of the directionality of time and uncertainty.
>
> **LIV**:
>
> LIV [4] is an extension of the VIP objective which enables interoperability with language with an additional vision-language contrastive alignment loss. In practice this is implemented similarly to VIP, with the addition of a CLIP infoNCE [5] alignment loss. Similar approaches to connecting language and visual representations have been used in several recent works [6,7,8].
>
> **Inference via Interpolation**:
>
> Inference via Interpolation [9] shows how a form of time-contrastive representation learning can enable analytic, explicit subgoal inference by ''interpolating'' in the representation space. This requires several geometric assumptions: uniformity of the representation marginals [10] and the assumption that the state and goal encoders differ by a linear transform. The advantage of TRA is that we don't need to explicitly infer subgoals to compose behaviors, which avoids computational cost and compounding errors.
>
> We will revise the paper to highlight these differences in our related work section.

---

> > ### Author Response · Authors · 2024-12-04
> > **References**
> >
> > [1] Ghugare, R. et al., 2024. ''Closing the Gap Between TD Learning and Supervised Learning—a Generalisation Point of View.'' *ICLR*
> >
> > [2] Park, S. et al., 2024. ''OGBench: Benchmarking Offline Goal-Conditioned RL.'' arXiv:2410.20092
> >
> > [3] Ma, Y. J. et al., 2023. ''VIP: Towards Universal Visual Reward and Representation via Value-Implicit Pre-Training.'' *ICLR*
> >
> > [4] Ma, Y. J. et al., 2023. ''LIV: Language-Image Representations and Rewards for Robotic Control.'' *ICML*
> >
> > [5] Radford, A. et al., 2021. ''Learning Transferable Visual Models From Natural Language Supervision.'' *ICML*
> >
> > [6] Jang, E. et al., 2021. ''BC-Z: Zero-Shot Task Generalization With Robotic Imitation Learning.'' *CoRL*
> >
> > [7] Xiao, T. et al., 2022. ''Robotic Skill Acquisition via Instruction Augmentation With Vision-Language Models.'' *RSS*
> >
> > [8] Myers, V. et al., 2023. ''Goal Representations for Instruction Following: A Semi-Supervised Language Interface to Control.'' *CoRL*
> >
> > [9] Eysenbach, B. et al., 2024. ''Inference via Interpolation: Contrastive Representations Provably Enable Planning and Inference.'' *NeurIPS*
> >
> > [10] Wang, T. and Isola, P., 2020. ''Understanding Contrastive Representation Learning Through Alignment and Uniformity on the Hypersphere.'' *ICML*

---

### Official Review · Reviewer_d83g · 2024-11-06

**Soundness:** 2
**Presentation:** 2
**Contribution:** 2
**Rating:** 3
**Confidence:** 4

**Summary:**

The paper investigates a novel approach called Temporal Representation Alignment (TRA) to enhance compositional generalization in robotic tasks, particularly for multi-step instruction following and goal-oriented tasks. TRA emphasizes learning representations that align temporally across different states, goals, and language instructions, which enables agents to perform complex, sequential tasks without additional planning. The method is tested on various robotic manipulation tasks in the BridgeData setup, showing significant improvements in compositional performance compared to other baseline methods like AWR and LCBC.

**Strengths:**

1. **Innovation in Representation Learning**: TRA introduces a creative approach to compositional generalization by structuring the alignment of temporal representations, which minimizes reliance on explicit planning or RL-based strategies.

2. **Zero-shot Compositionality**: TRA’s ability to generalize to unseen task combinations without retraining is a notable achievement, providing significant potential for scaling robotic applications in real-world, dynamically changing environments.

**Weaknesses:**

1. **Limited Scope of Task Complexity**: Although TRA shows compositionality, the tested tasks focus on relatively simple manipulations. More complex or multi-agent settings might challenge TRA's capabilities. (Most of them are pick-and-place)

2. **Dependence on Goal Representation Quality**: Success in tasks depends heavily on the quality and specificity of goal representations, which may require fine-tuning for certain task types.

3. **Missing Ablation Studies**: The authors have no ablation study on object-level instruction following and task-level instruction following since the work focuses on languange Instruction following. For example, "move the bell pepper to the bottom right of the table" v.s. "move the bell pepper to the bottom left of the table". It might overfit or replay the action sequence in the replay buffer.

**Questions:**

1. **Poor Baselines**: Why not choose diffusion policies for imitation learning baselines?

2. The alignment is too similar to the VIP method, why not give more explanations? (The author cites the VIP, but doesn't give any descriptions or comparisons)

3. It seems the font of the paper is wired. Have you chosen the right style?

---

> ### Author Response · Authors · 2024-11-24
> **Rebuttal**
>
> Thank you for your review and suggestions.
>
> It seems your primary concerns have to do with the evaluation tasks and baselines. We have run additional ablations on object-level reasoning as suggested. We discuss your remaining concerns below, highlighting revisions in the text with red. **Do these changes fully address your concerns**? We look forward to continuing the discussion.
>
> > No object-level ablation
>
> We have run additional object-level ablations, including the particular task suggested by the author.
>
> [**Table:** object-level ablation tasks](https://gcdnb.pbrd.co/images/UfZ0kvWXhaxd.png)
> | original task | new object-level ablation | original task TRA success rate (goal-conditioned) | new task TRA success rate (goal-conditioned) | original task TRA success rate (language-conditioned) | new task TRA success rate (language-conditioned) |
> | ----------------------------------- | ----------------------------------- | ------------------------------------------------- | -------------------------------------------- | ----------------------------------------------------- | ------------------------------------------------ |
> | *move bell pepper then sweep right* | *sweep right then move bell pepper* | 0.60 ± 0.2 | 0.90 ± 0.1 | 0.50 ± 0.2 | 0.70 ± 0.1 |
> | *corn on plate then sushi in pot* | *sushi in pot then corn in plate* | 0.30 ± 0.1 | 0.20 ± 0.1 | 0.7 ± 0.2 | 0.60 ± 0.2 |
> | *sweep towels right* | *sweep towels left* | 0.70 ± 0.1 | 0.50 ± 0.2 | 0.8 ± 0.1 | 0.70 ± 0.1 |
>
>
> The success rates for the new, reordered versions of the tasks do not differ from the original tasks significantly, showing the model is not just overfitting / memorizing sequences of events from the training data.
>
> > poor Baselines: Why not choose diffusion policies for imitation learning baselines?
>
> The precise parameterization of our policy head is fairly orthogonal to the main contribution of our work, which is studying how the *task representations* enable compositionality.
> We picked our current ResNet (+FiLM conditioning [1]) backbone to be consistent with past work using the BridgeData v2 [2], which has found similar architectures to be most effective.
>
> We have added discussion to our “Limitations and future work” section to address this.
>
> > too similar to VIP, why not give more explanations?
>
> Indeed, there are tight connections between our work and past approaches that use self-supervised representation learning for robotics.
> We would like to emphasize that our representations and analysis tackle the problem of compositionality, which is distinct from the benefits to representation learning studied in papers like VIP [3], LIV [4], or Voltron [5]. We have added more discussion on these methods to the related work section.
>
> > Limited evaluation tasks … more complex or multi-agent settings might challenge TRA's capabilities
>
> We agree that more complex settings are an exciting future direction. Our primary contribution in this paper is to show that a structured representation learning approach can provide an end-to-end robot learning algorithm for enabling a new property (compositionality) when used with an existing standard robot learning dataset (Bridge v2). We believe the scope of current evaluations on new robot manipulation tasks within the Bridge scenes is in line with prior work published at ICLR in similar settings [6,7]. We have added discussion to the future work section to suggest these future areas of exploration.
>
> > Dependence on Goal Representation Quality: Success in tasks depends heavily on the quality and specificity of goal representations, which may require fine-tuning for certain task types
>
> We have added discussion on this limitation to the related work section, which is also a practical limitation for any representation learning objective. **Does this address your concern?**
>
> > It seems the font of the paper is weird. Have you chosen the right style?
>
> We were using a computer modern font (MLModern) since the ICLR style files actually don’t specify a font. The revision has switched to Times New Roman. Does this address your concern?
>
> ### References
>
> [1] Perez, E. et al., 2018. "FiLM: Visual Reasoning With a General Conditioning Layer." *AAAI*
>
> [2] Walke, H. et al., 2023. "BridgeData V2: A Dataset for Robot Learning at Scale." *CoRL*
>
> [3] Ma, Y. et al., 2023. "VIP: Towards Universal Visual Reward and Representation via Value-Implicit Pre-Training." *ICLR*
>
> [4] Ma, Y. et al., 2023. "LIV: Language-Image Representations and Rewards for Robotic Control." *ICML*
>
> [5] Karamcheti, S. et al., 2023. "Language-Driven Representation Learning for Robotics." *ICRA*
>
> [6] Black, K. et al., 2024. "Zero-Shot Robotic Manipulation With Pre-Trained Image-Editing Diffusion Models." *ICLR*
>
> [7] Zheng, C. et al., 2024. "Stabilizing Contrastive RL: Techniques for Robotic Goal Reaching From Offline Data." *ICLR*

---

> ### Comment · Area_Chair_9yuw · 2024-11-27
> **Rebuttal Response Requested**
>
> Dear Reviewer, Do you mind letting the authors know if their rebuttal has addressed any of your concerns and questions? Thanks!

---

> > ### Author Response · Authors · 2024-12-03
> > **Follow-up**
> >
> > Dear Reviewer,
> >
> > We have made substantial revisions based on your feedback, as well as run the [additional real-world ablations](https://gcdnb.pbrd.co/images/UfZ0kvWXhaxd.png) you requested. **Do these revisions address your concerns?** If you have any remaining reservations, please inform us so we can make any needed revisions and/or clarifications.
> >
> > Thank you,
> >
> > The authors

---

### Meta-Review · Area_Chair_9yuw · 2024-12-17

**Metareview:**

**Summary** This paper addresses compositional generalization in goal-conditioned behavior cloning.  The authors propose a method called Temporal Representation Alignment (TRA), a self-supervised loss term which allows learning of representations which allow complex multi-step skills to be learned from demonstrations which do not show the full skill.

**Strengths** Reviewers broadly agreed the method was novel and showed strong empirical results versus baselines in a real-world experimental setting.  It was also noted that TRA appeared easy to use.

**Weaknesses** The greatest concern shared among all the reviewers was that the evaluation was not sufficient.  There was disagreement about whether the baselines and ablation were adequate, but the author response helped to alleviate these concerns. More critically, the evaluation tasks come from a single dataset, cannot be easily reproduced, and are potentially too simple. Another shared concern was the lack of theoretical motivation for TRA, which the authors largely addressed in their rebuttal and revision by adding explanations and a theory section. While similarity with past methods was discussed, I believe the authors made it clear their method is sufficiently different from previous methods.

**Conclusion** The scores for the paper are borderline and tend towards reject.  While the paper seems to have significant promise and strong empirical results and to have improved as a result of the rebuttal-revision phase, concerns with the comprehensiveness of the evaluation remain and should be addressed in a future submission.

**Additional Comments On Reviewer Discussion:**

4vMh gave clear review and gave the authors actionable feedback to improve their paper. The authors revised the paper in response including a new theory section to address the concerns of theoretical motivation. 4vMh increased their score to 5, but declined to give an accept rating due to outstanding concerns on the empirical evaluation.  d83g did respond to the rebuttals. I think the authors did a reasonable job of answering some of d83g concerns, but probably d83g would not have gone all the way to accept.

---

### Decision · Program_Chairs · 2025-01-22

Reject